# Checkpoint Kinase 1 Inhibitor Combined with Low Dose Hydroxyurea Promotes ATM-Activated NF-κB-Dependent Pro-Inflammatory Chemokine Expression in Melanomas

**DOI:** 10.3390/cancers17111817

**Published:** 2025-05-29

**Authors:** Nicole Lisa Li-Ann Goh, Nur Jannah Abdul Rahim, Rituparna Bhatt, Si En Ong, Khai Yee Lim, Anastasia Gandini, Zhen Zeng, Snehlata Kumari, Brian Gabrielli

**Affiliations:** 1Mater Research Institute, The University of Queensland, Brisbane, QLD 4072, Australia; nicolelisagoh@gmail.com (N.L.L.-A.G.); sien.ong@uq.net.au (S.E.O.); jenny.zeng@mater.uq.edu.au (Z.Z.); 2Frazer Institute, Dermatology Research Centre, Faculty of Health, Medicine and Behavioral Sciences, The University of Queensland, Brisbane, QLD 4072, Australia; s.kumari@uq.edu.au

**Keywords:** ATM, chemokine, NF-κB, replication stress

## Abstract

Tumours avoid detection by the body’s immune system through a range of mechanisms. An anti-cancer treatment that can alter this tumour-induced immune avoidance could enhance the ability of the patient’s own immune system to detect and respond appropriately to the tumour. Here, we show that a novel treatment that directly kills melanoma also upregulates chemokines, agents that can attract immune cells into the tumour to enhance an anti-tumour immune response. The mechanism by which this treatment increases these chemokines is also determined and involves a response to damage caused by the treatment that is also responsible for the tumour cell killing by this treatment. This study demonstrates that the new treatment can not only directly and effectively kill melanoma cells but also promote signals that will encourage immune cells to recognise the tumour.

## 1. Introduction

The tumour microenvironment (TME) contains tumour cells, immune and immunosuppressive cells, blood vessels, extracellular matrix, fibroblasts, and signalling molecules [1]. Anti-tumour immune responses are led by lymphocytes, including cytotoxic CD8^+^ T cells, CD4^+^ helper T cells, and natural killer (NK) cells, with the help of pro-inflammatory macrophages (M1) and dendritic cells. T cells and NK cells can be recruited to the TME via chemokines secreted by tumour cells or other immune cells [2,3,4,5], where they are referred to as tumour infiltrating lymphocytes (TILs). Tumour cells can also modify immune cell phenotypes and functions towards immunosuppression and promote tolerance to tumour antigens for immune evasion [1]. The immunosuppressive cells commonly found in the TME include regulatory T cells (Tregs), tumour-associated macrophages (TAMs), and myeloid-derived suppressor cells (MDSCs) [6]. Enhancing the anti-tumour immune response by reducing the immune suppression in the TME is, therefore, essential for the long-term responses to any anti-tumour treatment.

This can occur through multiple mechanisms, including expression of the dendritic cells chemoattractant XCL1, which increases tumour antigenicity to prime T cell responses [7], and secretion of T-cell recruiting chemokines such as CCL2, CCL5, CXCL9, and CXCL10 to attract cytotoxic CD8^+^ T cells to tumour sites [8].

Many cytokines and chemokines have dual, opposing roles and can have pro-inflammatory or anti-inflammatory functions depending on context and which immune cells are present [2]. Tregs and MDSCs reduce the number and/or activity of TILs in the TME [5]. Cancers recruit Tregs to the TME by expressing CCL5 and CXCL12, differentiating naïve CD4^+^ T cells, or converting Th17/CD8^+^ cells into Tregs via TGF-β to evade the immune system [3]. MDSCs are also recruited by cytokines/chemokines such as VEGF, TGF-β, CCL2, and CXCL12 to establish an immunosuppressive TME [9]. Tumour-associated macrophages (TAMs) are attracted to the TME through cytokines and chemokines, including VEGF, CSF-1, TGF-β, CCL2, CCL7, and CCL8. TAMs induce angiogenesis and can contribute to tumour metastasis, and higher densities of TAMs are associated with poorer clinical outcomes [3,10]. An immunosuppressive TME impedes the effectiveness of cancer therapy, making it crucial for novel therapies to modify the TME to a pro-inflammatory state to promote anti-tumour immune responses.

Chemotherapies that induce high levels of DNA damage can promote cytokine/chemokine expression in tumour cells [11]. This is often reported to be due to the activation of cGAS-STING by the presence of cytosolic DNA that is detected by cGAS and converted to cGAMP, which in turn activates STING [12]. STING activates TANK-binding kinase 1 (TBK1), activating IFN regulatory factor 3 (IRF3) and nuclear factor κB (NF-κB) that translocates into the nucleus driving transcription [12,13]. This promotes the expression of pro-inflammatory type-I interferons and cytokines such as tumour necrosis factor (TNF) and interleukin 6 (IL-6) [14]. However, other pathways respond to DNA damage to increase cytokine/chemokine gene expression, including NF-κB [15].

In previous studies, a combination of checkpoint kinase 1 inhibitor (CHK1i) and low-dose hydroxyurea (LDHU) has been shown to have tumour-selective toxicity, targeting tumours with elevated levels of replication stress without significant toxicity in normal tissue [16,17]. This was associated with increased expression of pro-inflammatory cytokines/chemokines in vivo, although the mechanism regulating this was not determined. Here, we investigate the mechanism regulating this treatment-induced chemokine expression in melanomas.

## 2. Materials and Methods

### 2.1. Cell Lines

The panel of human melanoma cell lines were originally sourced from Professor Nicholas Hayward QIMR-Berghofer Institute, Brisbane, Queensland, Australia [18]. The cells were cultured in Dulbecco′s Modified Eagle′s Medium (DMEM) (Sigma-Aldrich, St. Louis, MO, USA), 10% heat-inactivated Fetal Bovine Serum (Gibco, Waltham, MA, USA), GlutaMAX (Gibco), 20 mM HEPES (Sigma-Aldrich), and 1 mM Sodium Pyruvate (Sigma-Aldrich) with Antibiotic-Antimycotic (Gibco). Cells were maintained at 37 °C and 5% CO_2_ and were mycoplasma free. The mutation state of the cell lines is shown in Appendix A.

### 2.2. IncuCyte Assays

Cells were seeded in a 96-well plate in 100 µL media. Cells were acclimated for 24–48 h in a 37 °C and 5% CO_2_ incubator until 70% confluency before treating with 1 µM SRA737 + 0.2 mM hydroxyurea (HU), and 125 nM of SYTOX Green Fluorescence Dye (Thermofisher Scientific, Waltham, MA, USA) was added 30 min prior to imaging to visualise cell death. Cells were imaged for 72 h at 4 h intervals with 10× magnification using Sartorius Incucyte S3 Live Cell Analysis System (Essence Bioscience, Ann Arbor, MI, USA) at 37 °C and 5% CO_2_. Incucyte Base Analysis Software version 3.1.0 (Essence Bioscience) was used to perform segmentation and cell-by-cell analysis for cell counting and percentage death.

### 2.3. Immunoblotting

Cells were harvested and washed twice with PBS; then, the cell pellet was stored at −80 °C until required. Cell pellets were lysed in lysis buffer (NETN; 0.5% Nonidet P40, 100 mM NaCl, 20 mM Tris-Cl, and 2 mM EDTA), pH 8, 0.1% sodium dodecyl sulphate (SDS), 25 mM β-glycerophosphate, 25 mM sodium fluoride 0.1 mM sodium orthovanadate, Protease Inhibitor Cocktail (Sigma Aldrich), and 1 mM phenylmethylsulfonyl fluoride. Protein concentrations were normalised. Proteins were resolved on 4–20% SDS-polyacrylamide Mini-PROTEAN TGX (Bio-Rad, South Granville NSW, Australia) precast gels and then transferred onto 0.2 μM polyvinylidene difluoride (PVDF) membrane. Membranes were probed with the antibodies below, then detected using the appropriate horseradish peroxidase labelled secondary antibody and chemiluminescence detection. Primary antibodies were purchased from; phospho-Histone H2A.X (γH2AX) (Ser139/Tyr142) #5438 Cell Signaling Technology (Danvers, MA, USA), phospho-RPA2 Ser4/8 BL-165-5F1Bethyl Laboratories, phospho-ATM (Ser1981) #4526 Cell Signaling Technology, phospho-NF-κB p65 (RelA) (Ser536) #3033Cell Signaling Technology, NF-κB p65 (RelA) #8242Cell Signaling Technology, Histone H3 #9715Cell Signaling Technology, ATM GTX70103 GeneTex (Irvine, CA, USA).

### 2.4. RT-qPCR

Cells were treated for 24 h, then harvested using cell scrapers; media was removed before adding 1 mL of TRIzol reagent (Invitrogen, Waltham, MA, USA, #15596026). Total RNA was extracted from all cell lines using TRIzol. DNase 1 Kit (Invitrogen #18047019) was used according to the manufacturer’s instructions to remove DNA impurities from extracted RNA. RNA concentration and purity were assessed using Thermo Scientific NanoDrop One/OneC using the absorbance value ratio A260/A280 (1.8–2.0) and A260/A230 (>2.0), respectively. cDNA synthesis was performed using the High-Capacity cDNA Reverse Transcription Kit (Applied Biosystems, Waltham, MA, USA, #4368814), including RNaseOUT recombinant ribonuclease inhibitor (Invitrogen #10777019) according to the manufacturer’s instructions. Reverse transcription was performed using the Eppendorf Mastercycler pro PCR System (Fisher Scientific, Pittsburgh, PA, USA) with 2 µg of RNA. Predesigned KiCqStart primers (MERCK, Bayswater, VIC, Australia) were used for RT-qPCR, listed in Appendix A. PCR was performed using QuantStudio 7 Flex Real-Time PCR System (Thermo Fisher) in a 384-well plate. Melt curve analysis was performed to confirm specificity. For analysis of results, relative expression (ΔCt) values were normalised against the housekeeping gene YWHAZ (ΔCt = CtTarget Gene—CtYWHAZ), then log-transformed (2^−ΔCt^) for visual representation as fold change on a log10 *y*-axis.

### 2.5. Cytokine/Chemokine Bead Array

LEGENDplex Human Essential Immune Response Panel Cat#740930 (Australian Biosearch, Wangara, WA Australia), was used to assess the concentration of 13 cytokines/chemokines present in cell culture supernatants after treatment with SRA737 + LDHU was performed as per manufacturer’s instructions. Samples were analysed on a CytoFLEX S Flow Cytometer (Beckman Coulter, Lane Cove, Australia). Data were analysed using the provided LEGENDplex Data Analysis Software version 4 (BioLegend, San Diego, CA, USA).

### 2.6. Immunofluorescence

A2058 cells were seeded on coverslips in a 6-well plate with a density of 3 × 10^5^ cells/well. Cells were acclimated for 24 h until 70% confluency before respective treatments were added. Cells were fixed with 4% paraformaldehyde for 15 min at 37 °C, then washed and permeabilisation with 0.1% Triton X-100, blocked and immunostained for NF-κB p65 (RelA) #8242 Cell Signaling Technology using an Alexa488 labelled secondary antibody #A-11034 ThermoFisher and DAPI. Cells were mounted onto microscope slides and were imaged using an Olympus BX63 (Olympus Life Science, Waltham, MA, USA) microscope at 20× magnification. Cell segmentation and quantification of fluorescence intensity were performed using CellProfiler version 4.2.6 for Windows, www.cellprofiler.org.

### 2.7. Quantitative Analysis

All graphs and statistical analyses were performed using GraphPad Prism version 10.3.1 for Windows, GraphPad Software, Boston, MA, USA, www.graphpad.com. For statistical methods used, see individual figure legends.

## 3. Results

Treatment with the combination of 1 mM SRA737 and 0.2 mM HU promoted replication stress, indicated by the phosphorylation of RPA2 at Ser4/8, increased DNA damage indicated by increased γH2AX, and increased cell death (Figure 1A) as we have observed previously using a different CHK1 inhibitor [16]. It also promoted extensive cell killing in all the cell lines used here (Figure 1B; Appendix A).

DNA damage has been reported to also promote chemokine expression in many cancer types; in many cases, this is reported to be through the cGAS-STING pathway [19]. However, cGAS-STING is commonly down-regulated in many cancers, as demonstrated in the Cancer Cell Line Encyclopedia (CCLE) dataset, with approximately 35% of all cancer cell lines expressing low levels (<1 TPM) of cGAS and/or STING [20] (Figure 2A) and reported to be commonly inactivated in melanoma and other cancers [17,21]. The lack of typical cGAS-STING-induced type I interferon expression in response to SRA737 + low dose HU treatment in vivo [17] also suggested that this pathway was not responsible for the pro-inflammatory chemokine expression observed in two syngeneic melanoma models with this treatment (Figure 2B). In vitro treatment of a panel of human melanoma cell lines with SRA737 + low dose HU also resulted in increased expression of the pro-inflammatory chemokines CCL2, CCL5, CXCL8, CXCL10, IL6, XCL1, and TNF, and had modest effects on VEGFA and TGFB1 although the magnitude of expression was different for each cell line (Figure 2C). The changes in gene expression correlated with increased levels of the secreted chemokines, although little free TGFβ was detected (Appendix A).

It was previously reported that SRA737 + LDHU did not activate the cGAS-STING pathway in melanoma cell lines despite increased cytosolic DNA [17], suggesting other transcriptional pathways were likely to be involved. The majority of chemokines that showed increased expression with SRA737 + LDHU treatment were known NF-κB targets [22]. To investigate the potential involvement of NF-κB, the nuclear localisation of NF-κB RelA/p65 subunit was assessed, and the nuclear/cytoplasmic ratio of RelA was found to be significantly increased in treated cells (Figure 3A,B), indicating NF-κB activation was assessed.

The activation of NF-κB was confirmed using an inhibitor of NF-κB activation BI-605906, which blocked the SRA737 + LDHU-induced increased expression of CCL2, CCL5, CXCL8, CXCL10, IL6, and TNF to at least basal levels, although the NF-κB inhibitor also reduced the basal level of many of these chemokines to undetectable levels. The inhibitor had no effect on VEGFA or TGFB1, neither of which are regulated by NF-κB (Figure 4).

The increased DNA damage suggested the activation of the DNA damage response signalling through Ataxia telangiectasia mutated (ATM), and ATM was activated in response to SRA737 + LDHU in all cell lines (Figure 5A). ATM signalling in response to DNA damage has previously been shown to activate NF-κB [23], and inhibition of ATM using the inhibitor AZD1390 blocked the increase in RelA/p65 Ser536 phosphorylation observed with SRA737 + LDHU treatment to a similar level as the NF-κB inhibitor alone (Figure 5B). It also blocked the SRA737 + LDHU-induced increased expression of the NF-κB-regulated genes but had no effect on either VEGFA or TGFB1 (Figure 5C).

The same effect was observed with a second ATM inhibitor, KU-55933 (Appendix A). Although the NF-κB and ATM inhibitors effectively blocked the NF-κB-dependent chemokine expression, they did not affect either cell proliferation as single agents or cell killing by SRA737 + LDHU (Figure 6). This indicates that the NF-κB activation and downstream chemokines expression were not involved in the cell-killing response to treatment.

## 4. Discussion

This study showed that SRA737 + LDHU treatment of melanoma cells that is sufficient to kill a high proportion of cells also upregulates the expression of pro-inflammatory cytokines and chemokines cells, including *CCL2*, *CCL5*, *CXCL8*, *CXCL10*, *IL-6*, *TNF*, and *XCL1*, and immunosuppressive *VEGFA* and *TGFB1* before significant cell death occurs. The pro-inflammatory cytokines/chemokines have the potential to enhance the tumour’s immune microenvironment. Increased recruitment of NK cells and macrophages and activation of T cells was observed with this treatment in vivo [17], indicating that this pro-inflammatory program is a functional response. However, the increased expression of immunosuppressive *VEGFA* and *TGFB1* that are not regulated by NF-κB can oppose this pro-inflammatory effect. VEGFA can recruit suppressive immune cells such as TAMs, Tregs, and MDSCs, although it may also increase vascular permeability, thus improving drug delivery of SRA737 + LDHU to tumours in vivo [24]. TGF-β can promote malignancy [25] and immunosuppression by converting CD4^+^ T cells to Tregs [26]}. Here, we report that *TGFB1* is expressed at high levels in all the melanoma cell lines assessed, and SRA737 + LDHU treatment triggered a modest increase. However, there was little evidence of free TGF-β being secreted, suggesting the factors required for full maturation and release of free TGF-β are not present in melanomas [27]. In vivo, there is evidence of increased TGF-β following SRA737 + LDHU treatment, along with an increase in immunosuppressive FoxP3 + Tregs in the TME [17].

This study demonstrates the ability of SRA737 + LDHU to activate the NF-κB signalling pathway and is responsible for the upregulation of pro-inflammatory cytokines and chemokines observed. It is activated by ATM, a DNA damage sensor activated in response to the presence of the level of DNA double-strand breaks induced by replication fork collapse induced by CHK1 inhibition [28]. ATM regulates NEMO to, in turn, release IKKb to activate NF-κB [23]. NF-κB also appears to be responsible for much of the basal expression of *CCL2*, *CCL5,* and *TNF*, possibly a consequence of upregulated NF-κB activity, a common feature of melanoma [29]. CHK1 activated in response to replication stress can also phosphorylate RelA/p65 at Thr505, resulting in the inhibition of NF-κB-dependent anti-apoptotic gene regulation [30], but does not affect pro-inflammatory cytokine expression [31]. The effect of ATM on NF-κB activation is specific as other components involved in DNA damage response, such as DNA-PK and ATR, are not activators of NF-κB [15].

The upregulation of non-NF-κB targets suggests other transcriptional responses occur with SRA737 + LDHU. *XCL1* can be regulated by p53 [32], but with p53 commonly defective in melanomas, p63 and p73 may be responsible for regulating its transcription [33]. *TGFB1* and *VEGFA* can be regulated by the Activation Protein 1 (AP-1) complex (c-Jun/c-Fos) [34], which can also regulate the expression of *CXCL8* [35] and *IL-6* [36]. Since the MAPK pathway, which regulates AP-1 [37], is commonly overactive in melanoma [38], this may explain the high baseline expression of these cytokines/chemokines. The mechanism by which *TGFB1* and *VEGFA* with SRA737 + LDHU treatment is at present not known. However, blocking this pathway could reduce the immunosuppressive effects of these cytokines, enhancing the tumour’s immune microenvironment.

## 5. Conclusions

In conclusion, we provided evidence that SRA737 + LDHU activates NF-κB in human melanoma cells through the canonical pathway activated by the DNA damage response via ATM. This results in the upregulation of genes encoding for pro-inflammatory cytokines and chemokines, following a consistent general pattern across a variety of human melanoma cell lines independent of the degree of cell killing elicited by SRA737 + LDHU. These cytokines and chemokines can modify the tumour microenvironment from an immunosuppressive state to an inflamed state, promoting an anti-tumour immune response through the recruitment of immune cells to the tumour site.

## Figures and Tables

**Figure 1 cancers-17-01817-f001:**
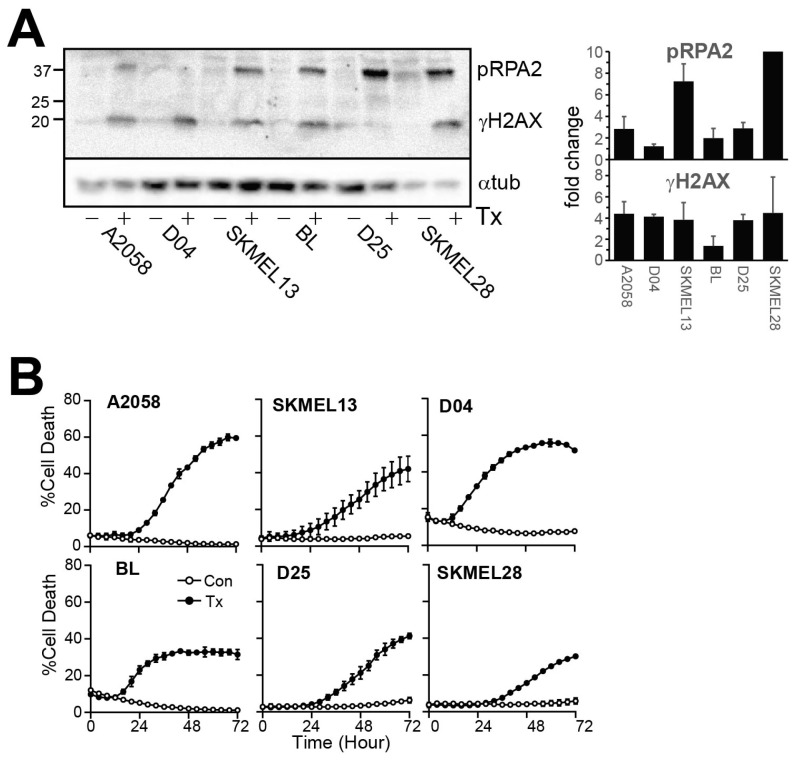
SRA737 + LDHU induces replication stress and DNA damage. (**A**) Cells were treated with or without 1 µM SRA737 + 0.2 mM HU and harvested at 24 h. Cell lysates were immunoblotted for γH2AX as a marker for DNA damage and RPA2 for replication stress. Histone H3 was used as a loading control. (**B**) The indicated melanoma cell lines were treated as in A and cells followed using Incucyte live cell imaging using SYTOX Green as a marker of cell death. The % cell death is shown. The data are the mean and SD from three replicates. The uncropped blots are shown in Appendix A.

**Figure 2 cancers-17-01817-f002:**
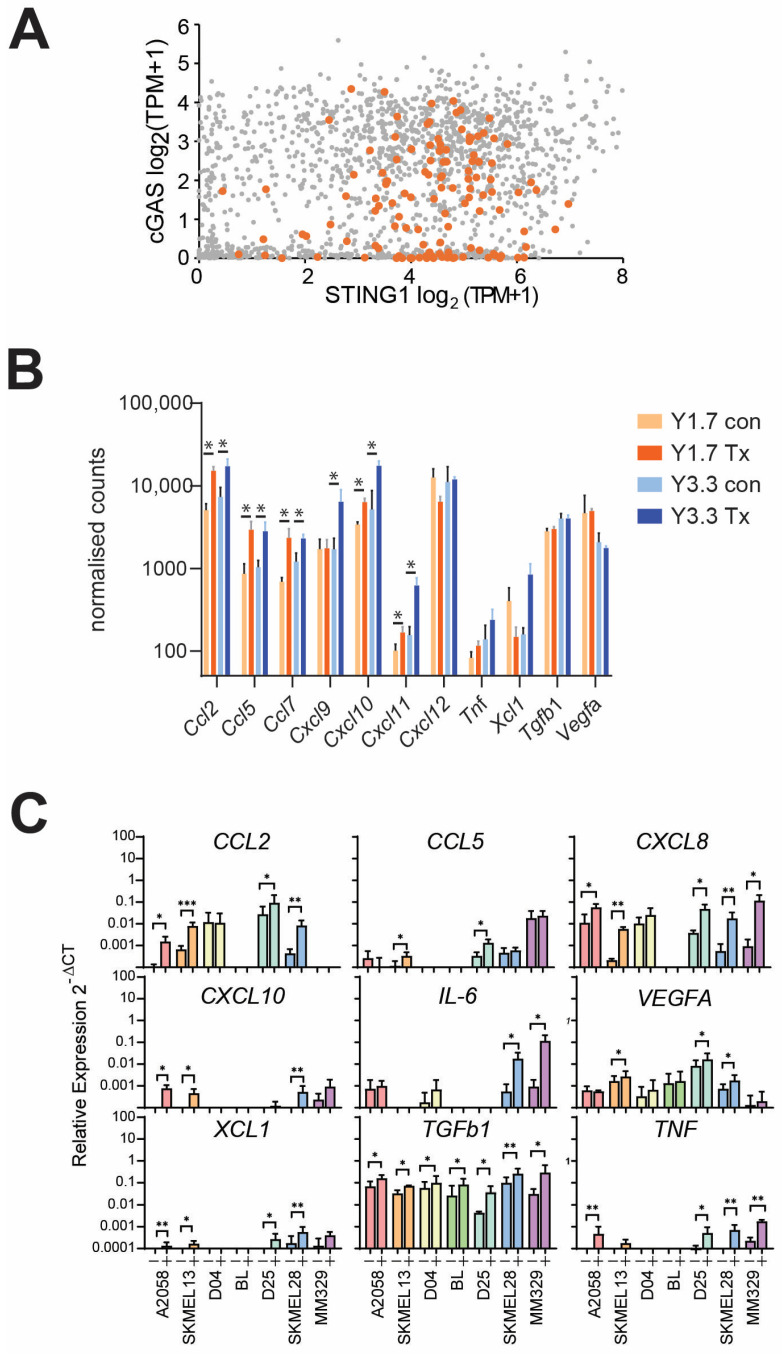
SRA737 + LDHU induces pro-inflammatory chemokine expression. (**A**) Gene expression data from 1673 human cancer cell lines for CGAS and STING from the CCLE. Melanoma cell lines are highlighted in orange. (**B**) Gene expression in murine melanoma (YUMMUV1.7 and YUMMUV3.3) was treated with and without SRA737 + LDHU in vivo for the indicated chemokines. The data are from 3–4 individual mice. (**C**) The melanoma cell lines were treated without or with SRA737 + LDHU (−, +) and harvested at 24 h for RNA extraction, followed by RT-qPCR. Data represents the mean with standard deviation error bars of 3 independent experiments, taking the mean of triplicate wells for each experiment. Multiple paired *t*-tests (parametric) were performed per gene on ΔCt values, followed by correction for multiple comparisons using the two-stage step-up method of Benjamini, Krieger, and Yekutieli false discovery rate (FDR) at a 5% threshold. Adjusted *p*-values are represented as * *p* < 0.05, ** *p* < 0.01, *** *p* < 0.001. The colours identify each cell line.

**Figure 3 cancers-17-01817-f003:**
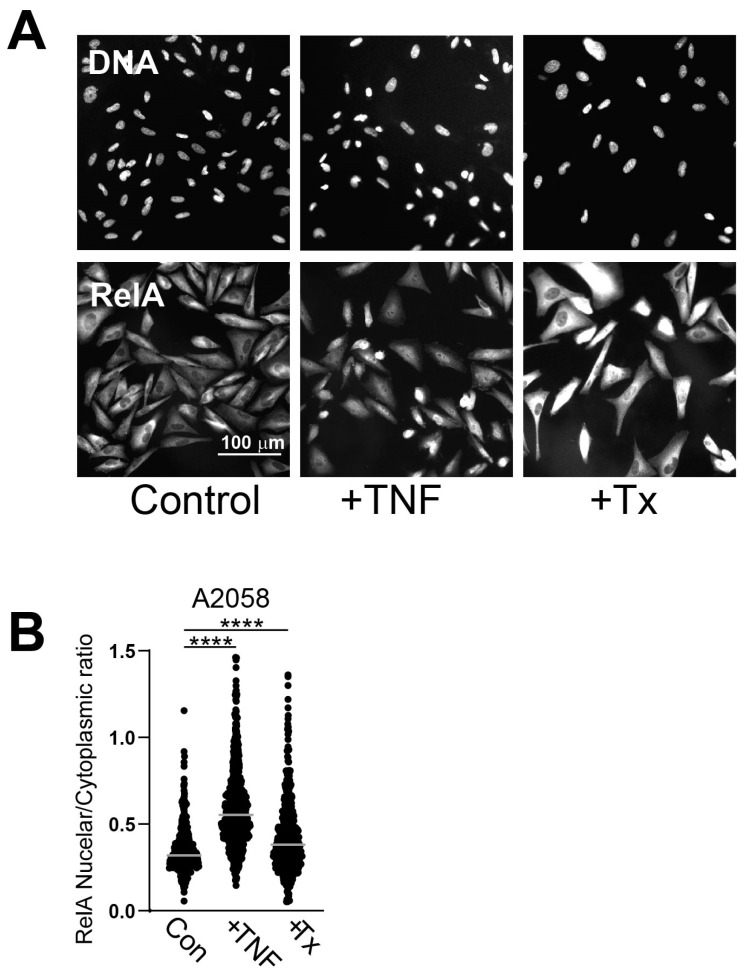
SRA737 + LDHU activates and drives nuclear localisation of NF-κB. (**A**) A2058 cells were treated with TNF-α for 1 h or SRA737 + LDHU for 24 h before fixing and labelling. Nuclei were stained with DAPI and RelA. Images were taken using Olympus BX63. The scale bar is 100 µm. (**B**) CellProfiler was used for cell and nucleus segmentation and quantification of the integrated intensity of RelA of nuclear and cytoplasmic RelA separately. The individual nuclear/cytoplasmic ratio of RelA is shown for 400–800 cells, and the mean is shown for each (light grey bar). One-way ANOVA was performed on ratios with *p* values were corrected using the Tukey test with a 0.05 (95% CI) threshold. Adjusted *p*-values are represented as **** *p* < 0.0001.

**Figure 4 cancers-17-01817-f004:**
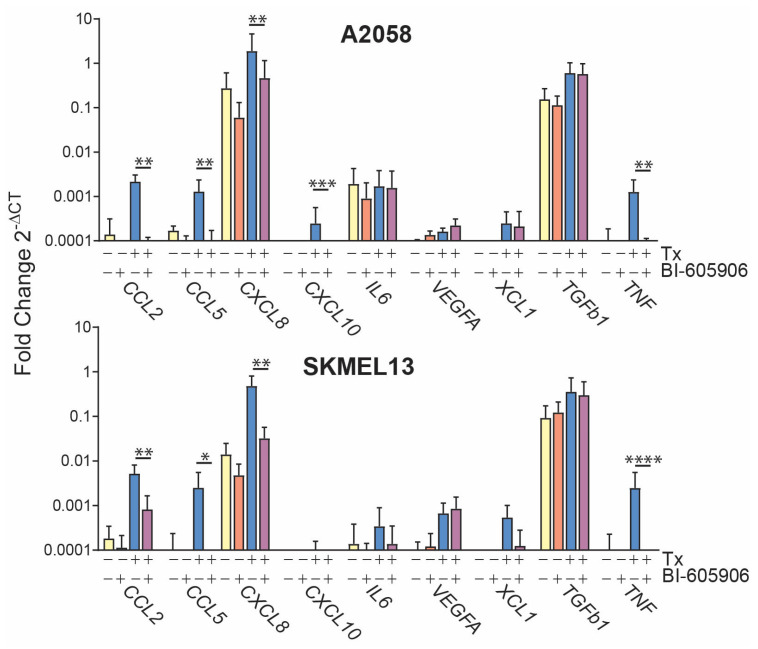
SRA737 + LDHU activates NF-κB-dependent chemokine expression. A2058 and SKMEL13 were treated with 20 µM BI-605906 (NF-κB inhibitor) for 45 min prior to treatment with 1 µM SRA737 + 0.2 mM HU and harvested at 24 h for RNA extraction, followed by RT-qPCR. Data represent the mean with standard deviation error bars and are representative of 3 independent experiments, taking the mean of triplicate wells for each experiment. Two-way ANOVA was performed on ΔCt values with multiple comparisons of biological replicate means of each treatment group. *p*-values were corrected using the Tukey test with a 0.05 (95% CI) threshold. Adjusted *p*-values are represented as * *p* < 0.05, ** *p* < 0.01, *** *p* < 0.001, and **** *p* < 0.0001. The colours identify each cell line.

**Figure 5 cancers-17-01817-f005:**
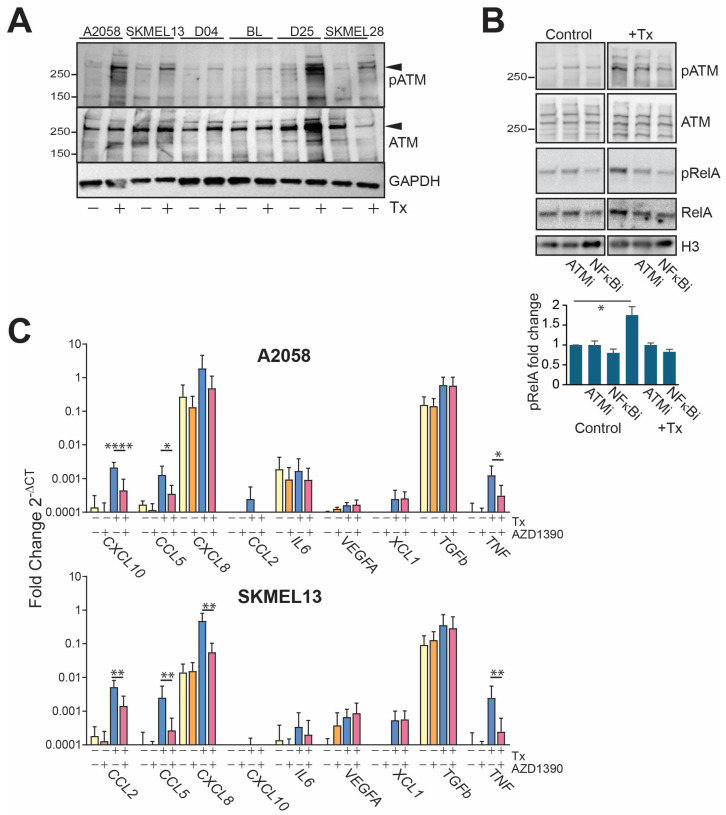
SRA737 + LDHU activates ATM, which in turn activates NF-κB-dependent chemokine expression. (**A**) The indicated melanoma cell lines were treated with 1 µM SRA737 + 0.2 mM HU for 24 h, and cells were harvested for immunoblotting of the indicated proteins. (**B**) A2058 cells were treated with 100 nM AZD1390 or 20 µM BI-605906, with 1 µM SRA737 + 0.2 mM HU for 24 h, and lysates immunoblotted for the indicated proteins. Histone H3 was used as a loading control. The levels of pRelA relative to the untreated control from 3 independent experiments are shown. (**C**) A2058 and SKMEL13 were treated with ATM inhibitor 100 nM AZD1390 with 1 µM SRA737 + 0.2 mM HU for 24 h and harvested for RNA extraction, followed by RT-qPCR. Data represent the mean with standard deviation error bars and is representative of 3 independent experiments (*n* = 3), taking the mean of triplicate wells for each experiment. Two-way ANOVA was performed on ΔCt values with multiple comparisons of biological replicate means of each treatment group. *p*-values were corrected using the Tukey test with a 0.05 (95% CI) threshold. Adjusted *p*-values are represented * *p* < 0.05, ** *p* < 0.01, and **** *p* < 0.0001. Only relevant *p*-values are displayed. The colours identify each cell line. The uncropped blots are shown in Appendix A.

**Figure 6 cancers-17-01817-f006:**
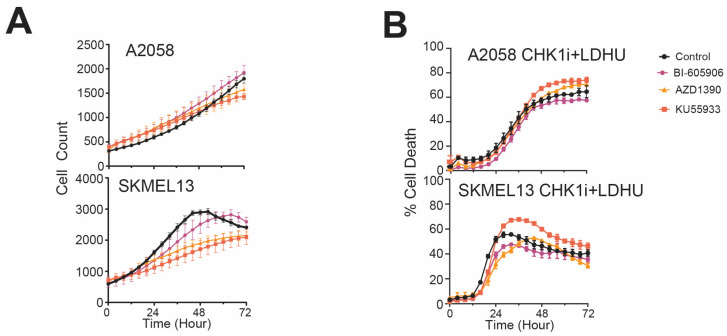
Reduction in cell death was not observed with ATM, and NF-κB signalling inhibitors A2058 and SKMEL13 cells were pre-incubated for 45 min with NF-κB signalling inhibitor, BI-605906 (20 μM), or ATM inhibitors KU55933 (10 μM) and AZD1390 (1 μM) with or without 1 μM SRA737+ 0.2 mM HU. Cells were then imaged using IncuCyte for 72 h using SYTOX Green as a marker of cell death. (**A**) The cell number is shown for the NF-κB and ATM inhibitors. (**B**) The percent cell death for the cell treated with SRA737 + LDHU and NF-κB and ATM inhibitors.

## Data Availability

The original contributions presented in this study are included in the article/Appendix A. Further inquiries can be directed to the corresponding author.

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
