# Peer review of "Checkpoint Kinase 1 Inhibitor Combined with Low Dose Hydroxyurea Promotes ATM-Activated NF-κB-Dependent Pro-Inflammatory Chemokine Expression in Melanomas"

_cancers, 2025, doi:10.3390/cancers17111817_

Round 1

Reviewer 1 Report

Comments and Suggestions for Authors

In the present manuscript entitled: CHK1 inhibitor combined with low dose hydroxyurea promote ATM activated NF-kB dependent pro-inflammatory chemokine expression in melanomas, the authors focused on investigating the potential impact of SRA737+ LDHU as an anti-tumor agent against melanoma, using cell lines. The data presented is new and the manuscript was clearly presented, however, several points should be seriously taken in consideration for the following raisons:

  • The authors should respect the use of abbreviations throughout the manuscript. The abbreviation must be defined at the first appear in the text and then the authors should use the abbreviations throughout the manuscript.
  • The keywords must be alphabetically re-arranged and should be correctly revised.
  • The introduction section was poorly written and should be improved with more recent references.
  • In the Materials and methods section, the authors stated in the Immunofluorescence that “and immunostained using an appropriately labelled secondary antibody and DAPI” the authors should add the information about the secondary Abs. Additionally, the statistical analysis was incorrectly written and must be re-written in details.
  • In figure 1A, the authors presented the protein bands of one representative experiment. The authors must normalize the phosphorylation levels and the expression levels of RPA2 and H2AX on the loading control and accumulated data from three experiments must be presented in bars under the protein bands. Similarly, the same comments were observed for figure 2A and B.
  • In figure 4, the authors measured the expression levels of cytokines/chemokines at the mRNA levels, and this scientifically is not correct and must be measured at the protein levels using ELISA, IHC, WB, or flow cytometry.
  • The discussion section was poorly written and must be improved.

Author Response

In the present manuscript entitled: CHK1 inhibitor combined with low dose hydroxyurea promote ATM activated NF-kB dependent pro-inflammatory chemokine expression in melanomas, the authors focused on investigating the potential impact of SRA737+ LDHU as an anti-tumor agent against melanoma, using cell lines. The data presented is new and the manuscript was clearly presented, however, several points should be seriously taken in consideration for the following raisons:

The authors should respect the use of abbreviations throughout the manuscript. The abbreviation must be defined at the first appear in the text and then the authors should use the abbreviations throughout the manuscript.

We have amended the abbreviations list as requested.

The keywords must be alphabetically re-arranged and should be correctly revised.

We have amended the keywords as requested.

The introduction section was poorly written and should be improved with more recent references.

We have added more references and revised the text.  The oldest reference is 2017 which is still less than 10 years old.

In the Materials and methods section, the authors stated in the Immunofluorescence that “and immunostained using an appropriately labelled secondary antibody and DAPI” the authors should add the information about the secondary Abs. Additionally, the statistical analysis was incorrectly written and must be re-written in details.

We have amended the Materials and Methods section about the secondary antibody used  as requested.  The statistical methods used for each of the figures is described in detail in the figure legends for each of figure.  As the methods deal specifically with the data shown we consider it more beneficial for the reader to have the statistical methods detailed at the point of use rather than in the Methods section.

In figure 1A, the authors presented the protein bands of one representative experiment. The authors must normalize the phosphorylation levels and the expression levels of RPA2 and H2AX on the loading control and accumulated data from three experiments must be presented in bars under the protein bands. Similarly, the same comments were observed for figure 2A and B.

The data in figure 1A is confirmatory data to demonstrate that the CHK1 inhibitor SRA737 when combined with 0.2 mM HU produce the same effects as a previous CHK1 inhibitor GDC-0575 used in a previous paper (Oo et al., 2019). The text has been amended to refer to this paper.  We have also modified the immunoblots to show the phosphoRPA2 Ser4/8 band in place of the total RPA2 immunoblot and provided the quantification requested by both reviewers.

In figure 4, the authors measured the expression levels of cytokines/chemokines at the mRNA levels, and this scientifically is not correct and must be measured at the protein levels using ELISA, IHC, WB, or flow cytometry.

We respectfully disagree with the review on this point.  This figure examines the regulation of chemokine gene expression by NF-kB and therefore measurement of the mRNA levels by RT-qPCR is the correct method of analysing this.  Your suggestions to measure the protein production and/or secretion, will be the sum of the changes in gene expression, protein synthesis, protein stability and secretion.  Although it is likely that changes in protein levels as we have measured with our treatment for some of the chemokines and shown in Supplementary Figure S2, will be reduced by the inhibition of NF-kB, the direct effect is the changes in mRNA level. 

The discussion section was poorly written and must be improved.

We have revised the Discussion.

  1. Proctor M, Gonzalez Cruz JL, Daignault-Mill SM, Veitch M, Zeng B, Ehmann A, et al. Targeting Replication Stress Using CHK1 Inhibitor Promotes Innate and NKT Cell Immune Responses and Tumour Regression. Cancers (Basel). 2021;13(15).

Reviewer 2 Report

Comments and Suggestions for Authors

The study “CHK1 inhibitor combined with low dose hydroxyurea promote ATM activated NF-kB dependent pro-inflammatory chemokine expression in melanomas” focuses on the combinatorial treatment of CHK1 inhibitor and low dose hydroxyurea and its impact on the production of pro-inflammatory cytokines and chemokines. CHK1 inhibitor and low dose hydroxyurea treatment was shown to increase pro-inflammatory cytokines through ATM-NF-kB signaling in melanoma cells. Interestingly, introduction of NF-kB and ATM inhibitors on top of this combinatorial treatment did not alter proliferation nor cell killing. The authors speculates that the upregulation of chemokines and cytokines might evoke its anti-tumor activity through remodelling of tumor microenvironment instead.

Overall, this is an interesting and relevant study that addresses current pre-clinical treatment strategy in melanoma (doi: 10.1111/pcmr.13120, doi: 10.1002/1878-0261.12497). The manuscript is well written and it incorporates relevant literature. However, I have several suggestion points that I would like to make to strengthen the paper into a publishable form (in order of figures):

  • Figure 1A: Given H3 is used as loading control, there should be approximately similar protein level for compared conditions. If the level is far from similar, then authors should quantify their immunoblots.
  • Figure 2A: As the study focuses on melanoma, it might be helpful to also show the correlation in melanoma cell lines.
  • Figure 2B: I think it might be useful to present the data of gene expression in similar manner as Figure 2C.
  • General Figure 2: I think the authors should perform experiments to rule out role of cGAS-STING activation in addition to CCLE database.
  • Figure 3B: Instead of mean intensity, I would suggest the authors to quantify nuclear:cytoplasmic RelA as measure of RelA activation
  • Figure 5A: Legend stated treatment with AZD1390 or BI-239 605906, but the panel only shows Tx (Chk1i + LDHU) instead
  • Figure 6: SKMEL13 CHK1i + LDHU a drop is observed past 24 hours, can the authors speculate behind this phenomenon?

Author Response

The study “CHK1 inhibitor combined with low dose hydroxyurea promote ATM activated NF-kB dependent pro-inflammatory chemokine expression in melanomas” focuses on the combinatorial treatment of CHK1 inhibitor and low dose hydroxyurea and its impact on the production of pro-inflammatory cytokines and chemokines. CHK1 inhibitor and low dose hydroxyurea treatment was shown to increase pro-inflammatory cytokines through ATM-NF-kB signaling in melanoma cells. Interestingly, introduction of NF-kB and ATM inhibitors on top of this combinatorial treatment did not alter proliferation nor cell killing. The authors speculates that the upregulation of chemokines and cytokines might evoke its anti-tumor activity through remodelling of tumor microenvironment instead.

Overall, this is an interesting and relevant study that addresses current pre-clinical treatment strategy in melanoma (doi: 10.1111/pcmr.13120, doi: 10.1002/1878-0261.12497). The manuscript is well written and it incorporates relevant literature. However, I have several suggestion points that I would like to make to strengthen the paper into a publishable form (in order of figures):

Figure 1A: Given H3 is used as loading control, there should be approximately similar protein level for compared conditions. If the level is far from similar, then authors should quantify their immunoblots.

We have revised this figure to use the phosphoRPA2 Ser4/8 antibody that is a better indicator of replication stress.  The immunoblots have the quantified as requested. 

Figure 2A: As the study focuses on melanoma, it might be helpful to also show the correlation in melanoma cell lines.

We have changed this figure as requested.

Figure 2B: I think it might be useful to present the data of gene expression in similar manner as Figure 2C.

We have changed this figure as requested.

General Figure 2: I think the authors should perform experiments to rule out role of cGAS-STING activation in addition to CCLE database.

We have previously shown that cGAS-STING pathway components are either missing in many melanoma cell lines, and where present we detected no pathway activation with CHK1i+LDHU treatment (1).

Figure 3B: Instead of mean intensity, I would suggest the authors to quantify nuclear:cytoplasmic RelA as measure of RelA activation.

We have changed this to Nuclear/cytoplasmic ratio as requested.

Figure 5A: Legend stated treatment with AZD1390 or BI-239 605906, but the panel only shows Tx (Chk1i + LDHU) instead.

The legend has been amended.  Panel A had not been mentioned in the original legend.

Figure 6: SKMEL13 CHK1i + LDHU a drop is observed past 24 hours, can the authors speculate behind this phenomenon?

The data is reported as % cell death i.e. the % cell that are SYTOX Green positive at that time point.  As the cell die they will detach from plate and float out of the field of view at the bottom of the plate. So at later time points there are very few cells attached and a proportion of those few remain SYTOX negative. So while the % SYTOX positive appears to reduce, it is the % of a very small and reducing number indicated by the total cell count.

  1. Proctor M, Gonzalez Cruz JL, Daignault-Mill SM, Veitch M, Zeng B, Ehmann A, et al. Targeting Replication Stress Using CHK1 Inhibitor Promotes Innate and NKT Cell Immune Responses and Tumour Regression. Cancers (Basel). 2021;13(15).